# Assessment of Knowledge on Metal Trace Element Concentrations and Metallothionein Biomarkers in Cetaceans

**DOI:** 10.3390/toxics11050454

**Published:** 2023-05-12

**Authors:** Vincent Leignel, Louis Pillot, Marcela Silvia Gerpe, Florence Caurant

**Affiliations:** 1Laboratoire BIOSSE, Le Mans Université, Avenue O Messiaen, 72000 Le Mans, Francelouis.pillot.etu@univ-lemans.fr (L.P.); 2Universidad Nacional de Mar del Plata, Funes 3350, Mar del Plata CP. 7600, Argentina; msgerpe@mdp.edu.ar; 3Consejo Nacional de Investigaciones Científicas y Técnicas (CONICET), Godoy Cruz 2290, Argentina; 4Observatoire Pelagis, UAR3462 La Rochelle University,5 all. De l’océan, 17000 La Rochelle, France; 5Centre d’Études Biologiques de Chizé (CEBC), UMR 7372 CNRS-La Rochelle Université, 79360 Villiers en Bois, France

**Keywords:** cetaceans, pollution, metallothioneins, gene, mRNA

## Abstract

Cetaceans are recognized as bioindicators of pollution in oceans. These marine mammals are final trophic chain consumers and easily accumulate pollutants. For example, metals are abundant in oceans and commonly found in the cetacean tissues. Metallothioneins (MTs) are small non-enzyme proteins involved in metal cell regulation and are essential in many cellular processes (cell proliferation, redox balance, etc.). Thus, the MT levels and the concentrations of metals in cetacean tissue are positively correlated. Four types of metallothioneins (MT1, 2, 3, and 4) are found in mammals, which may have a distinct expression in tissues. Surprisingly, only a few genes or mRNA-encoding metallothioneins are characterized in cetaceans; molecular studies are focused on MT quantification, using biochemical methods. Thus, we characterized, in transcriptomic and genomic data, more than 200 complete sequences of metallothioneins (mt1, 2, 3, and 4) in cetacean species to study their structural variability and to propose to the scientific research community *Mt* genes dataset to develop in future molecular approaches which will study the four types of metallothioneins in diversified organs (brain, gonad, intestine, kidney, stomach, etc.).

## 1. Introduction

The large number of chemical compounds (medical compounds, Metal Trace Elements, pesticides, plastics, etc.) occur in the marine ecosystem often biodegrade slowly [1] They may come from natural and anthropogenic activities and may be concentrated through the food chain. Dolphins and whales are the final consumers of trophic networks in the marine ecosystem. Some cetaceans filter their food (small crustaceans and fish), whereas others are predators of cephalopods and fish. Thus, a large diversity of pollutants accumulates by biomagnification in cetacean’s tissues, mainly from ingested food, which affects their health [2,3]. A majority of the pollutants are endocrine disturbers or generate cellular oxidative stress [4]. To respond to these negative effects, the organisms synthesize many molecules, playing a role in detoxification processes. Metals are one of the most abundant pollutants in oceans and seas. The organisms, accumulating high metal concentration, synthesize an important metallothionein quantity, and they are non-enzymatic proteins involved in metal detoxification.

### 1.1. Metals Are Ubiquitous Pollutants in Cetaceans

The metals are highly present in cetacean tissues, and their accumulation appears proportional to the levels in the environment and their prey [5], suggesting that dolphins and whales may be considered to be sentinel (quantitative bioindicator) to reflect the quality of the marine environment [6,7,8,9]. However, many ecological and physiological factors modulate the chance to recover the metals in cetaceans: specie, age, sex, body size, nutritive conditions, and diet [10,11,12].

Non-essential metals (Element Trace Metals, ETMs) may have embryotoxic, nephrotoxic, neurotoxic, and reprotoxic effects, an inducer of immune depression, inducing DNA damage, teratogenic effects, cell proliferation, and oxidative stress [8,13,14,15,16]. Nevertheless, essential metal elements protect against ETM effects. This protective effect could be because essential metals (e.g., Zn) are inducers of the synthesis of metallothioneins (MTs), which are involved in metal detoxification [17]. The metal concentrations in cetaceans are mainly estimated in the kidney and liver because these organs are, respectively, involved in immune response, biotransformation of toxic compounds, and renal filtration; however, some studies are also focused on metal levels in muscle [7,18,19,20,21,22,23,24,25,26,27,28,29]. Unfortunately, it is not possible to compare the metal contaminations determined in distinct cetacean species, because they were collected in different geographical zones and years. In this case, it could be interesting in the future to investigate metal contaminations in more tissues, such as the brain and the digestive tube (esophagus, stomach, intestine, spleen, or the skin), as well as in different species collected in the same locality.

### 1.2. Metallothionein, a Biomarker in Response to Metal Contaminations

Many publications that studied the metal content in the tissues of cetaceans are focused on the metallothionein concentration because their cellular synthesis is correlated to metal accumulation. MTs’ induction has been considered one of the most important detoxification processes against metal toxicity and is also involved in the regulation of apoptosis and redox balance equilibrium [8,30]. Thus, MTs are considered to be a molecular bioindicator of metal exposure and are used commonly as a tool for biomonitoring programs.

Metallothioneins (MTs) are small non-enzymatic proteins (61–68 amino acids, 6–7 kDa) that are extremely rich in cysteine amino-acids (>30%) [31], which are organized in alternating Cys-Cys, Cys-X-Cys, and Cys-X-X-Cys (X being an amino-acid other than cysteine). Cysteine is implicated in metal complexation [32,33,34]. The MT binding affinity is metal-dependent [35,36]. In mammals, four types of metallothioneins are found: MT1, MT2, MT3, and MT4 [37]. The MT1 and MT2 are expressed in most tissues, whereas MT3 and MT4 (minor isoforms) are expressed in specified tissues [38]. MT3, considered to be a growth-inhibiting factor, is mainly expressed in Central Nervous System but it may be detected in the heart, kidney, and reproductive organs [39]. MT4 is specific to stratified tissues such as the oral epithelium, esophagus, stomach, and skin. Thus, MT1 and MT2 are involved in metal detoxification, homeostasis, and transport, whereas MT3 and MT4 functions are probably involved in tissue differentiation. It is suggested that the metallothionein family evolved by successive duplication genes. Duplicated copies may have accepted an accelerated rate of mutation, under selective pressure, promoting increased gene diversity and following subfunctionalization protein [40].

Mammalian MT is composed of two domains separated by a linker. The alpha domain (C-terminal) incorporates four metal cations bound with eleven cysteine residues, and a beta domain (N terminal) includes three metal cations bound to nine cysteines [41]. The biosynthesis of MTs depends mainly on metal accumulation in tissues, even if it may also be produced in response to various other regulator factors, such as glucocorticoids and temperature, depending on the activation of distinct enhancer regions in the promotor [42,43].

### 1.3. Characterization of Metallothioneins in Cetaceans

The first description of MTs in cetaceans was made by Ridlington et al. [44], who identified metal-binding proteins in the liver of *Physeter macrocephalus* (sperm whale). In 1986, Kwohn et al. [45] identified two isoforms of MTs (6.8 kDa), including 20–21 cysteine residues (32.7–33.3%), from the kidneys of *Stenella coeruleoalba* (Striped dolphin). These proteins were revealed as being close to MT1 and MT2 from the horse. Das et al. [46,47] confirmed the existence of MT1 and MT2 in the kidney and liver of *Delphinus delphis*, *Lagenorhynchus albirostris*, *L. acutus*, *Phocoena phocoena*, and *Physeter macrocephalus*. Mehra and Bremmer [48] indicated that the MT2 expression may be more prolonged, whereas the MT1 degradation is faster. Parallelly, Caurant et al. [49] showed that mercury (Hg) accumulation in pilot whales (*Globicephala melas*) was not correlated to metallothionein-like proteins in the liver because it was mainly found in the insoluble fraction. Ikemoto et al. [50] also revealed that the MTs that were identified in hepatic cytosol of *Phocoenoides dalli* (Dall’s porpoises) were not bound to silver (Ag), but a linear relationship existed between the Cd, Cu, and Zn content and the MTs synthesis. Das et al. [51] and Pedrero et al. [52] confirmed that Hg was mainly found to be complexed to high-molecular-weight proteins (HMWPs), probably as the HgSe form (tiemannite), and not to the MTs. Pollizi et al. [53] investigated the metallothioneins’ induction during ontogeny (fetus, calves, juveniles, and adult) of the coastal Franciscana dolphin *Pontoporia blainvillei*. They revealed that fetal MT concentrations were higher than in the mothers. The fetal period is characterized by a high metabolic rate during development and growth, and this may explain why high metal concentration is mainly in the liver of the fetus. For example, it may be possible that there is a metal transfer from mother to fetus. Càceres-Saez et al. [54], in relation to the MT/metal ratio, showed that MT/Cd was higher in the liver of *Cephalorhynchus commersonii*, whereas MT/Hg and MT/Ag were higher in the kidney, revealing a differential tissues accumulation.

Surprisingly, the majority of publications that were cited previously evaluated the MT concentration in tissues by using the spectrophotometric methods (absorbance at 412 nm) described by Elmman [55] or Viarengo et al. [56]. Unfortunately, these spectrophotometric methods did not allow for the discrimination of distinct MT isoforms. Their molecular approach can be explained by the fact that only a few nucleotide sequences of *Mts* have been well characterized from the genomes and transcriptomes of cetaceans yet. Liu et al. [57] published an innovative study focused on the metallothionein genes. They characterized the *Mt2* and *Mt4* alleles associated with metal levels in dolphin tissues (kidney, liver, and muscle). They identified two polymorphic sites only in the *Mt4* gene which seemed to be associated with Cd, Hg, Mn, and Zn content in *Neophocaena asiaeorientalis*’s tissues. Many chromosomes, scaffolds, contig, and transcriptomes of cetaceans are available in nucleotide international databases, but any gene annotation is performed.

Our main objective in this study was to constitute an *Mt* genes dataset to give the opportunity to the scientist community to develop future precise molecular approaches which can be used to evaluate the *Mt* expression for all genes (*Mt1*, *2*, *3*, and *4*) in many tissues (such as the brain, esophagus, gonad, heart, skin, stomach, and intestine, which are not integrated into metal content analyses yet). Thus, we decided to identify the metallothionein sequences in all genomic fragments (scaffold, contig, and read), cDNA, and transcriptomes of cetaceans available in international databases.

## 2. Material and Methods

### 2.1. Characterization of Metallothionein Sequences inside Available Transcriptomes and Genomes of Cetaceans

In the international database, only fifty MTs sequences were submitted, constituting a disparate dataset (mainly MT1 and MT4), including many MT1-E pseudogene sequences. This limited dataset explains why the metallothionein studies in cetaceans are mainly focused on the MT biosynthesis protein. The typical *Mt* gene structure includes three exons and two introns in mammals. Two first exons encode to the beta domain of the protein, while the third exon encodes to the alpha domain [58,59].

We screened the Nucleotide collection (nr/nt), Whole-Genome Shotgun Contigs (WGSs), Expressed Sequence Tags (ESTs), and Transcriptome Shotgun Assembly (TSAs) available at NCBI, using the BLASTn program (https://blast.ncbi.nlm.nih.gov/Blast.cgi accessed on 20–27 September 2022), selecting only the cetacean sequences. The intron localizations in genomic metallothionein sequences were determined by comparison with the mRNA of *Mt* from mammals, and the relevance of encoding sequences was verified by an *in silico* translation (https://web.expasy.org/translate/ accessed on 20–27 September 2022) and the blast program. The proteins obtained were compared, using the BLASTp program, to other MT sequences of the international database.

### 2.2. Phylogenetic Analysis of Metallothioneins in Cetaceans

We aligned the metallothionein dataset using the MAFFT algorithm with the default parameters (http://mafft.cbrc.jp/alignment/server/ accessed on 1–10 October 2022). Evolutionary analyses were conducted in MEGA XI (https://www.megasoftware.net/ accessed on 1–10 October 2022). The best evolutionary model for our dataset was determined, and the Maximum Likelihood method was applied [60,61]. A test of phylogeny used was bootstrap; only node values equal to 100 are shown in the figure.

## 3. Results

A total of more than 200 complete sequences were isolated from 26 species of cetaceans (dolphins and whales) included in the 13 families (*Balaenidae*, 2; *Balaenopteridae*, 4; *Delphinidae*, 6; *Eschrichtiidae*, 1; *Iniidae*, 1; *Kogiidae*, 1; *Lipotidae*, 1; *Monodontidae*, 2; *Phocoenidae*, 3; *Physeteridae*, 1; *Platanistidae*, 1; *Pontoporiidae*, 1; *Ziphiidae*, 2) (Table 1).

To show the total of Mt genes which were characterized in the cetacean species, we built a molecular phylogeny by using the mitochondrion sequences of the 26 species (accession numbers of mitochondrion were indicated in Table 1). The evolutionary history was inferred by using the Maximum Likelihood method and General Time Reversible model. The tree with the highest log likelihood (−129,411.30) is shown. The percentage of trees in which the associated taxa clustered together is shown next to the branches. Initial tree(s) for the heuristic search were obtained automatically by applying Neighbor-Joining and BioNJ algorithms to a matrix of pairwise distances estimated using the Maximum Composite Likelihood (MCL) approach and selecting the topology with superior log likelihood value. A discrete Gamma distribution was used to model the evolutionary rate differences among sites (five categories (+*G*, parameter = 1.2020)). The rate variation model allowed for some sites to be evolutionarily invariable ([+*I*], 45.31% sites). This analysis involved 27 nucleotide sequences because the mitochondrial genome of *Hippopotamus amphibius* (NC_000889) was used as an outgroup. Codon positions included were 1st+2nd+3rd+Noncoding. There was a total of 15,952 positions in the final dataset (Figure 1).

We built a phylogenetic tree based on the encoding nucleotide (mRNA, gene) sequences of metallothionein characterized in cetaceans, using also the MEGA XI (Maximum Likelihood method and Kimura two-parameter model and tree with the highest log likelihood: −2304.11, +*G*, parameter = 0.7943, 219 positions in the final dataset). This analysis allowed us to determine the cluster of *Mt* genes.

We showed that there is a unique copy of *Mt4*, *Mt3*, and *Mt2* genes in cetacean genomes but successive duplicated *Mt1* copies (*Mt1a*, *Mt1b*, and *Mt1c*). The length of the InterGenic Regions (IGRs) inside the metallothionein cluster (*Mt4-Mt3-Mt2-Mt1*) was calculated (Table 2). The IGR (*Mt4/Mt3*) is highest (19,583–36,109 bp). The IGR (*Mt3-Mt2*) ranges from 7129 to 7571 bp, IGR (*Mt2-Mt1*) from 2064 to 5310 bp, and the IGR between distinct *Mt1* copies varying approximately within 3000 bp (Table 2). This information is primordial to people whose genes amplify the successive Mt genes by PCR. To design specific primers for long PCR, people may report to Table 2, where they will find the accession number of the contig, scaffold or gene for each species where we identified the distinct *Mt* isoforms. The intron and exon sizes were also determined (Table 3 and Table 4). High stability of exon lengths was noted between the species and for each gene: Exon I (28–31 bp), Exon II (66 bp), and Exon III (92 bp), except for the *Mt3*, which showed the highest exon III (104–107 bp) (Table 4). The intron length was highly variable. The *Mt4* appeared to be the longest gene (±4500 bp).

Phylogenetic analyses based on 213 nucleotide metallothionein sequences (encoding part: ATG-TAA/TAG) identified in this study clearly showed four clusters (*Mt1*, *Mt2*, *Mt3*, and *Mt4*) (Figure 2). It is noted that the intron-free *Mt2* genes identified constitute a specific cluster, whereas the intron-free *Mt1* genes are dispatched (Figure 2). The isoforms *Mt1, Mt2*, and *Mt3* are more closed than *Mt4*. MT1 and MT2 are synthesized in many tissues, whereas MT3 is mainly mentioned in regard to the Central Nervous System and MT4 in stratified tissues. It is possible to suggest that these phylogenetic relationships may be explained by successive duplicates of the ancestral gene of metallothionein, which gave *Mt1* and *Mt2*, then *Mt3*, and, more recently, *Mt4*.

## 4. Conclusions

This study revealing the identification of more than 200 sequences of metallothioneins in genomes and transcriptomes sequences of 26 cetacean species constitutes a novel tool to develop a gene expression inside distinct tissues not used yet (brain, esophagus, gonad, heart, stomach, intestine, etc.) and in the skin. Now, using our indication, it is possible for people to design specific primers to develop a study of the metallothionein gene expression in cetaceans. It will increase our knowledge of the involvement of these molecular biomarkers in the detoxification responses of cetaceans against marine pollution. For example, we will analyze the gene expression of four *Mt* in distinct tissues (brain, intestine, kidney, and liver) of *Globicephala melas* to estimate if there is a differential response. Parallelly, another publication focused on the evolution of metallothionein in marine mammals, based on the structural analysis, positive selection events, and annotation errors of some *Mt* sequences available in the nucleotide database, will be written.

## Figures and Tables

**Figure 1 toxics-11-00454-f001:**
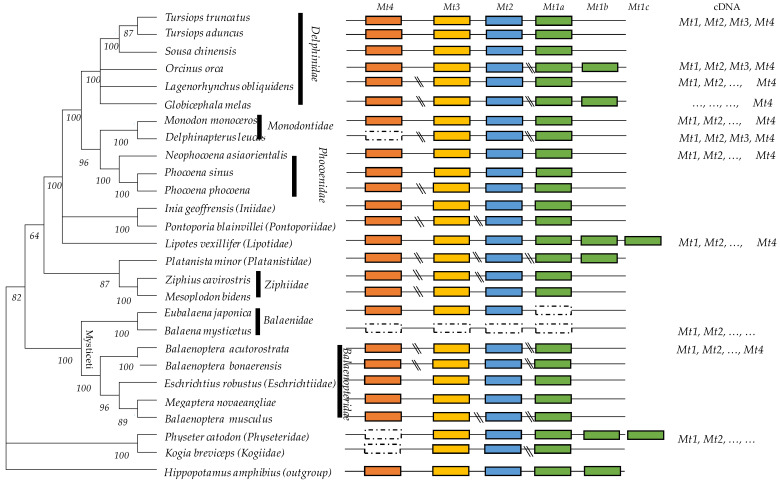
Genomic organization of metallothioneins in the phylogenetic tree of 26 cetacean species. Functional genes (*Mt1*, *Mt2*, *Mt3*, and *Mt4*) are indicated in colored rectangles. Phylogenomic analyses were based on mitogenome information and built using the MEGA XI (Maximum Likelihood method and General Time Reversible model, G + I parameters). Bootstrap values are indicated above the branches. The symbol “\\” indicated a large InterGenic Region.

**Figure 2 toxics-11-00454-f002:**
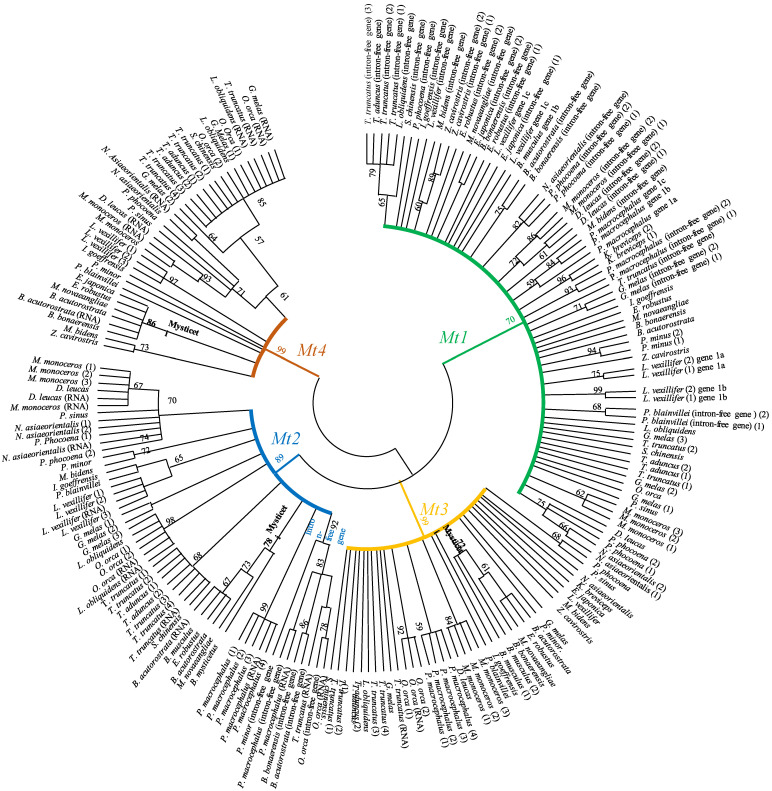
Phylogenetic tree of metallothioneins (*Mts*) in cetaceans. The relationships among the *Mts* genes are estimated using the Kimura 2-parameter model, a tree with the highest log likelihood: −2304.11, +*G*, parameter = 0.7943, 219 positions in the final dataset. Numbers above the nodes correspond to bootstrap values. Branches in green, blue, yellow, and brown indicate *Mt1*, *Mt2*, *Mt3*, and *Mt4*, respectively.

**Table 1 toxics-11-00454-t001:** Accession numbers of mitochondrion and metallothionein sequences of cetaceans isolated from NCBI databases and personal data.

Species	Family	Mitochondrion	A Cluster of *Mt* Genes (Contig)	Free-Intron *Mt* Gene	*Mt* Gene with Intron	*Mt* mRNA
*Balaena mysticetus*	*Balaenidae*	NC_005268				*Mt1*: SRR17645797, *Mt2*: AF022117
*Balaenoptera acutorostrata*	*Balaemopteridae*	NC_005271	*Mt3-Mt2*: ATDI01082660-661	*Mt1*: ATDI01132427, ATDI01131740, ATDI01081017, *Mt2*: ATDI01041776	*Mt1*: ATDI01082662 *Mt4*: ATDI01082657	*Mt1*: XM_007167588, XR_003622960, XR_003622964, XR_003622677, XM_007193604, XM_007193603, XM_007193605, XM_007193602, XM_007167587, XM_007167582, XM_007198029, XM_007178847, XM_007167583, XM_007164196, XM_007175023 *Mt2*: XM_007167584 *Mt4*: XM_028162920
*Balaenoptera bonaerensis*	*Balaenopteridae*	NC_006926	*Mt3-Mt2*: BAUQ01093692	*Mt1*: BAUQ01195848, BAUQ01613210 *Mt2*: BAUQ01307512	*Mt1*: BAUQ01180559 *Mt4*: BAUQ01234274	
*Balaenoptera musculus*	*Balaenopteridae*	MF409242	*Mt4-Mt3:* VNFD01000017	*Mt1:* VNFD01005770	*Mt1:* VNFD01001442 *Mt2*: VNFC01000015 *Mt3*: VNFC01000015	
*Delphinapterus leucas*	*Monodontidae*	NC_034236	*Mt3-Mt2-Mt1:* NQVZ01021645	*Mt1*: NQVZ01003203		*Mt1*: XM_022555910, XR_002642604 *Mt2*: XM_022555911 *Mt3*: GGBT01018098 *Mt4*: XM_022555909
*Eschrichtius robustus*	*Eschrichtiidae*	NC_005270	*Mt4-Mt3-Mt2-Mt1*: RJWN010001895	*Mt1*: RJWN010023894, RJWN010012658, NIPP01004159, NIPP01000460, RJWN010001847, NIPP01050325, RJWN010001516, NIPP01000414		
*Eubalaena japonica*	*Balaenidae*	NC_006931	*Mt4-Mt3-Mt2:* RJWP010002310	*Mt1*: RJWP010029515, RJWP010014455, RJWP011049311, RJWP010044123, RJWP010003859, RJWP010517844, RJWP010045258		
*Globicephala melas*	*Delphinidae*	NC_019441	*Mt3-Mt2:* SWEB01012070 *Mt1-Mt1*: SWEB01015162	*Mt1*: personal data	*Mt1*: personal data *Mt2*: personal data *Mt3*: personal data *Mt4*: personal data	*Mt4*: XM_030847937
*Inia geoffrensis*	*Iniidae*	NC_005276	*Mt4-Mt3-Mt2-Mt1:* RJWO010009779	*Mt1*: RJWO010024955, RJWO010005555		
*Kogia breviceps*	*Kogiidae*	NC_005272	*Mt3-Mt2:* RJWL010036575	*Mt1*: RJWL010001100	*Mt1*: RJWL010167408	
*Lagenorhynchus obliquidens*	*Delphinidae*	NC_035426	*Mt3-Mt2-Mt1:* RCWK01007239	*Mt1*: RCWK01003005	*Mt4*: RCWK01007238	*Mt1*: XM_027119875, XM_027092179, XR_003432150 XR_003433512, XR_003429008 *Mt2*: XM_027119873 *Mt4*: XM_027119872
*Lipotes vexillifer*	*Lipotidae*	NC_007629	*Mt4-Mt3*: AUPI01085919, *Mt4-Mt3-Mt2-Mt1-Mt1*: NW_006786802, *Mt2-Mt1-Mt1-Mt1*: AUPI01085920	*Mt1*: AUPI01016291, AUPI01032811	*Mt2*: AUPI01085920	*Mt1*: XM_007450307, XM_007459446, XM_007459445, XM_007459251 *Mt2*: XM_007459444 *Mt4:* XM_007459443
*Megaptera novaengliae*	*Balaenopteridae*	NC_006927	*Mt4-Mt3-Mt2-Mt1*: RYZJ01000704	*Mt1*: RYZJ01002277		
*Mesoplodon bidens*	*Ziphiidae*	NC_042218	*Mt3-Mt2-Mt1*: PVJJ010038290	*Mt1*: PVJJ010001248, PVJJ010035753, PVJJ010000716	*Mt4*: PVJJ010048532	
*Monodon monoceros*	*Monodontidae*	NC_005279	*Mt4-Mt3-Mt2-Mt1*: SIHG01006952, *Mt3-Mt2-Mt1:* PVJE01024091, PVJF01025398	*Mt1*: PVJF01009326, PVJE01004367, SIHG01006957, RWIC01000029		*Mt1*: XM_029210741, XM_022566857, XR_003793663, XM_029213332, XM_029213320, XM_029213311, XR_003792376, XM_029243937 *Mt2*: XM_029210730 *Mt4*: XM_029207165
*Neophocaena asiaeorientalis*	*Phocoenidae*	NC_026456	*Mt3-Mt2*: MKKW01002943	*Mt1*: XR_003002470, MKKW01009685, MKKW01050072	*Mt1*: MKKW01002942, NW_020172079 *Mt2*, *Mt4*: NW_020172079	*Mt1*: XM_024751546, XM_024752871 *Mt2*: XM_024751535 *Mt4*: XM_024751560
*Orcinus orca*	*Delphinidae*	NC_023889	*Mt3-Mt2*: ANOL02076608, *Mt1-Mt1*: ANOL02076611, *Mt4-Mt3-Mt2*: NW_004438720	*Mt1*: ANOL02015359, ANOL02053476, ANOL02005809 *Mt2*: ANOL02033101	*Mt4*: ANOL02076604	*Mt1*: XR_001119644, XR_001120057, XM_004286377 *Mt2*: XM_004272467, XM_004286376 *Mt3*: XM_004286375 *Mt4*: XM_004286374
*Phocoena phocoena*	*Phocoenidae*	NC_005280	*Mt3-Mt2-Mt1*: RJWQ010020171, *Mt2-Mt1*: PKGA01134162	*Mt1*: RJWQ010000213, PKGA01141836, RJWQ010001208, PKGA01000712	*Mt4*: RJWQ010008745	
*Phocoena sinus*	*Phocoenidae*	MZ772969	*Mt4-Mt3-Mt2-Mt1*: VOSU01000010			
*Physeter catodon*	*Physeteridae*	KU891394	*Mt3-Mt2*: PGGR02120841, *Mt1-Mt1-Mt1*: PGGR02120842, *Mt3-Mt2-Mt1*: AWZP01019177, *Mt3-Mt2*: UEMC01002060	*Mt1*: AWZP01108577, AWZP01061491, UEMC01000019, PGGR02098163, AWZP01094651, AWZP01036149 *Mt2*: AWZP01013965	*Mt2*: personal data *Mt3*: personal data	*Mt1*: XM_028487398, XM_028487397, XR_002892573, XM_024124849, XR_002890606, XR_002890985, XM_007111896, XM_007106395 *Mt2*: XR_002891953, XM_024124902, XR_002891953, XM_007104604
*Platanista minor*	*Platanistidae*	NC_005275	*Mt1-Mt1*: RJWK010077258	*Mt1*: RJWK010030898 RJWK010019970	*Mt1*: RJWK010077258 *Mt2*: RJWK010120550 *Mt3*: RJWK010071068 *Mt4*: RJWK010018570	
*Pontoporia blainvillei*	*Pontoporiidae*	NC_005277	*Mt2-Mt1*: RJWI010022586	*Mt1*: RJWI010118407	*Mt1*: RJWI010124881 *Mt3*: RJWI010018702 *Mt4*: RJWI010009362	
*Sousa chinensis*	*Delphinidae*	NC_012057	*Mt4-Mt3-Mt2-Mt1*: QWLN02017480	*Mt1*: QWLN02012546, QWLN02014060		
*Tursiops aduncus*	*Delphinidae*	KF570360	*Mt4-Mt3-Mt2-Mt1*: NCQN01002597	*Mt1*: NCQN01000091, NCQN01000487		
*Tursiops truncatus*	*Delphinidae*	EU557093	*Mt4-Mt3-Mt2-Mt1*: NW_004202941, QUXD02065780, *Mt4-Mt3-Mt2:* ABRN02426572	*Mt1*: ABRN02374863, ABRN02315981, ABRN02301451, QUXD02004646, QUXD02000953, QMGA01000002, MRVK01000157, QUXD02061382, QUXD02003404, QMGA01000469, MRVK01000730,	*Mt2*: QUXD02065780, ABRN02426572	*Mt1*: XM_019951660, XR_002175011, XR_002178769, XR_002172975, XM_004322331 *Mt2*: XM_004331916, XM_004322332 *Mt3*: XM004322333 *Mt4*: XM004322334
*Ziphius cavirostris*	*Ziphiidae*	KC776698	*Mt2-Mt1*: RJWS010029178	*Mt1*: RJWS010650073, RJWS011128550, RJWS010020051	*Mt3*: RJWS010262321 *Mt4*: RJWS010091044	

**Table 2 toxics-11-00454-t002:** Length (base pairs, bp) of InterGenic Region (IGR) between metallothionein genes (*Mt1*, *Mt2*, *Mt3*, and *Mt4*) in cetacean species.

Species	IGR (*Mt4-Mt3*)	IGR (*Mt3-Mt2*)	IGR (*Mt2-Mt1a*)	IGR (*Mt1a-Mt1b*)	IGR (*Mt1b-Mt1c*)
*Balaenoptera acutorostrata*		7129			
*Balaenoptera bonaerensis*		7137			
*Balaenoptera musculus*	21933				
*Delphinapterus leucas*		7173	5015		
*Eschrichtius robustus*	21903	7521	4739	3141	
*Eubalaena japonica*	22674	7171			
*Globicephala melas*		7202		3068	
*Inia geoffrensis*	19583	7232	5310	3084	
*Kogia breviceps*		7195			
*Lagenorhynchus obliquidens*		7184	5026		
*Lipotes vexillifer*	20328	7897	5309	3214	3926
*Megaptera novaengliae*	±22163	7175	4740	3083	
*Mesoplodon bidens*		7196	4994		
*Monodon monoceros*	27526	7159	5005		
*Neophocaena asiaeorientalis*		7201		3092	
*Orcinus orca*	±36109	7194		3055	
*Phocoena phocoena*		7198	5137	3091	
*Phocoena sinus*	27958	7196	5090		
*Physeter catodon*		7206	4448	2729	3088
*Platanista minor*			5083	3048	
*Pontoporia blainvillei*			2064		
*Sousa chinensis*	32782	7177	5016		
*Tursiops aduncus*	±34939	7182	5027		
*Tursiops truncatus*	23064	7186–7219	6269	3151	
*Ziphius cavirostris*			4955		
Average ± SD (bp)	25913.5 ± 5840.65	7236.25 ± 174.37	4895.71 ± 818.35	3068.72 ± 122.72	3507 ± 592.55
Min–Max (bp)	19,583–36,109	7129–7571	2064–5310	2729–3214	3088–3926

**Table 3 toxics-11-00454-t003:** Exon length (base pairs, bp) for metallothionein genes (*Mt1*, *Mt2*, *Mt3*, and *Mt4*) in cetacean species.

Species	*Mt1*			*Mt2*			*Mt3*			*Mt4*		
	Exon I	Exon II	Exon III	Exon I	Exon II	Exon III	Exon I	Exon II	Exon III	Exon I	Exon II	Exon III
*Balaenoptera acutorostrata*	28	66	92	28	66	92	31	66	107	31	66	92
*Balaenoptera bonaerensis*	28	66	92	28	66		31	66	107	31	66	92
*Balaenoptera musculus*	28	66	92	28	66	92	31	66	107	31	66	92
*Delphinapterus leucas*	28	66	92	28	66	92	31	66	107			
*Eschrichtius robustus*	28	66	92	28	66	92	31	66	107	31	66	92
*Eubalaena japonica*				28	66		31	66	107	31	66	92
*Globicephala melas*	28	66	92	28	66	92	31	66	107	31	66	92
*Inia geoffrensis*	28	66	92	28	66	92	31	66	107	31	66	92
*Kogia breviceps*	28	66	92	28	81		31	66	107			
*Lagenorhynchus obliquidens*	28	66	92	28	66	92	31	66	107	31	66	92
*Lipotes vexillifer*	28	66	92	28	66	92	31	66	107	31	66	92
*Megaptera novaengliae*	28	66	92	28	66	92	31	66	107	31	66	92
*Mesoplodon bidens*	28			28	66	92	31	66	104	31	66	92
*Monodon monoceros*	28	66	92	28	66	92	31	66	107	31	66	92
*Neophocaena asiaeorientalis*	28	66	92	28	66	92	31	66	107	31	66	92
*Orcinus orca*	28	66	92	28	66	92	31	66	107	31	66	92
*Phocoena phocoena*	28	66	92	28	66	92	31	66	107	31	66	92
*Phocoena sinus*	31	66	92	28	66	92	31	66	107	31	66	92
*Physeter catodon*	28	66	92	28	66	92	31	66	104			
*Platanista minor*	28	66	92	28	66	92	31	66	107	31	66	92
*Pontoporia blainvillei*	28	66		28	66	92	31	66	107	31	66	92
*Sousa chinensis*	28	66	92	28	66	92	31	66	107	31	66	92
*Tursiops aduncus*	28	66	92	28	66	92	31	66	107	31	66	92
*Tursiops truncatus*	28	66	92	28	66	92	31	66	107	31	66	92
*Ziphius cavirostris*	28	66	92	28		92	31	66	104	31	66	92
Min–Max (bp)	28–31	66	92	28	66–81	92	31	66	104–107	31	66	92

**Table 4 toxics-11-00454-t004:** Intron length (base pairs, bp) for Metallothionein genes (*Mt1*, *Mt2*, *Mt3*, and *Mt4*) in cetacean species.

Species	*Mt1(a)-mt1(b)-mt1(c)*		*Mt2*		*Mt3*		*Mt4*	
	Intron I	Intron II	Intron I	Intron II	Intron I	Intron II	Intron I	Intron II
*Balaenoptera acutorostrata*	611	351	295		175	854	1642	527
*Balaenoptera bonaerensis*	611	351	295		175	854	1646	527
*Balaenoptera musculus*	575(c)	737(c)	295	209	175	856	1666	527
*Delphinapterus leucas*	601	352	295	216	173	854		
*Eschrichtius robustus*	611	351	295	219	175	857	1671	527
*Eubalaena japonica*			295		175	855	1662	527
*Globicephala melas*	611(a)-578(b)	350(a)-710(b)	295	216	173	853	1171	526
*Inia geoffrensis*	611	344	291	215	175	849	1368	527
*Kogia breviceps*	577	351	295	221	181	854		
*Lagenorhynchus obliquidens*	611	350	295	216	173	851	1308	526
*Lipotes vexillifer*	608(a)-721(b)-578(c)	350(a)-753(b)-708(c)	295	214	175	853	1436	527
*Megaptera novaengliae*	611	351	295	205	175	854	1608	526
*Mesoplodon bidens*			294	215	175	861	1395	526
*Monodon monoceros*	612	355	295	216	173	854	1395	527
*Neophocaena asiaeorientalis*	610	350	295	217	173	850	1391	527
*Orcinus orca*	611	350	295	216	173	853	1309	526
*Phocoena phocoena*	610	350	295	217	173	849	1393	527
*Phocoena sinus*	601	352	295	217	173	849	1390	527
*Physeter macrocephalus*	611(a)-713(b)-716(c)	353(a)-359(b)-359(c)	295	216	176	857		
*Platanista minor*	594	356	297	224	176	803	1399	529
*Pontoporia blainvillei*	611		295	215	175	861	1357	520
*Sousa chinensis*	611	351	294	216	173	852	1308	527
*Tursiops aduncus*	611	350	295	216	173	852	1309	527
*Tursiops truncatus*	611(a)-612(b)	350(a)-714(b)	295	216	173	1098	1309	527
*Ziphius cavirostris*	608	351			175	861	1393	526
Average ± SD (bp)	607.30 ± 7.95(a) 682 ± 60.75(b) 611.75 ± 69.51(c)	351 ± 2.20(a) 608.67 ± 217.1(b) 628.50 ± 180.15(c)	294.85 ± 0.92	218.88 ± 14.13	174.38 ± 1.75	861.38 ± 49.40	1425.98 ± 142.38	548.130 ± 105
Min–Max (bp)	577–612(a) 612–721(b) 575–716(c)	344–356(a) 359–753(b) 359–737(c)	291–297	205–283	173–181	803–1098	1171–1671	520–1041

## Data Availability

All metallothionein sequences characterized in this study are available according to the accession number indicated in Table 1.

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
