# Peer review of "Assessment of Knowledge on Metal Trace Element Concentrations and Metallothionein Biomarkers in Cetaceans"

_toxics, 2023, doi:10.3390/toxics11050454_

Round 1

Reviewer 1 Report

toxics-2329882

In this manuscript, a study was conducted to investigate metallothioneins in cetaceans in terms of identification Mt gene sequences. In the transcriptomic and genomic data of cetacean species, the authors identified complete sequences of metallothioneins and proposed a Mt gene dataset to develop molecular probes to be used as biomarkers to investigate the presence of metallothioneins in diverse organs of cetaceans. This approach may provide a new tool for developing gene expression within distinct tissues.

This study contains novelties and proposes interesting tools to be used to design specific primers to study metallothionein gene expression in cetaceans. This approach will allow to understand the involvement of these molecular biomarkers in the detoxification responses of cetaceans against marine pollution.

The manuscript is presented as an article, with the results reported in figures 1 and 2, the tables cited in the text of the manuscript are not included. However, the introductory section divided into four paragraphs seems that of a review, with the description of the distribution of metallothioneins in the different organs of cetaceans and with the description of the properties of the different metallothioneins.

The Introduction section is too long and scattered and the purpose of the study may be lost.

The Materials and Methods section is not well defined, as is the Discussion section.

The manuscript contains interesting insights, however the study organization and methods are not clear and the whole message is not exhaustive.

Revisions

Line 39: ‘Some of Ttese’, please check;

lines 63-64 and 73: (Table 1) was mentioned in the text but it was not included;

line 76: ‘oesophagi’ change to ‘esophagus’;

line 102: ‘oral epithelia, esophagi,’ change to ‘oral epithelium, esophagus,’;

lines 227-229: Change the names of family to Italics;

line 229: (Table 2) was mentioned in the text but it was not included;

line 232: the link is not avalilable;

line 258: (Table 3) was mentioned in the text but it was not included;

line 264: (Tables 4 and 5) were mentioned in the text but they were not included;

line 288: ‘esophagi’ change to ‘esophagus’.

Author Response

1/ “The manuscript is presented as an article, with the results reported in figures 1 and 2, the tables cited in the text of the manuscript are not included.”

AUTHORS: We think that it was an error because figures and Tables are presented in the manuscript.

2/ “The Introduction section is too long and scattered and the purpose of the study may be lost. The Materials and Methods section is not well defined, as is the Discussion section.”

AUTHORS: We changed the organization of the manuscript and now there are distinct parts Introduction (best organized and short) and Materials & Methods.

3/ “Revision

Line 39: ‘Some of Ttese’, please check; AUTHORS: we changed

lines 63-64 and 73: (Table 1) was mentioned in the text but it was not included; AUTHORS: we verified

line 76: ‘oesophagi’ change to ‘oesophagus’; AUTHORS: we changed

line 102: ‘oral epithelia, esophagi,’ change to ‘oral epithelium, oesophagus,’ AUTHORS: we changed

lines 227-229: Change the names of the family to Italics; AUTHORS: we changed

line 229: (Table 2) was mentioned in the text but it was not included; AUTHORS: it was a mistake, we verified

line 258: (Table 3) was mentioned in the text but it was not included; AUTHORS: it was a mistake, we verified

line 264: (Tables 4 and 5) were mentioned in the text but they were not included; AUTHORS: it was a mistake, we verified

line 288: ‘esophagi’ change to ‘oesophagus’. AUTHORS: we changed

Reviewer 2 Report

The article is very interesting and thoughtful. The authors have written and structured a superb document that is based on very important research for environmental knowledge.
I am of the opinion that this article should be published and that only a few aspects to be corrected by the authors should be considered.
1) The figures are not clearly visible, the resolution of the image makes them difficult to read on some occasions. Revise all the figures.
2) The conclusions are a bit brief. Perhaps explain a little more, especially the contribution and future perspective of the results obtained in this research.

Author Response

1/ The figures are not clearly visible, the resolution of the image makes them difficult to read on some occasions. Revise all the figures. AUTHORS: We increased the size and quality of the figures.

2/ The conclusions are a bit brief. Perhaps explain a little more, especially the contribution and future perspective of the results obtained in this research. AUTHORS: We added some sentences to explain our project on the metallothionein of cetaceans.

Reviewer 3 Report

In their study, the author(s) identified more than 100 complete metallothionein sequences (mt1, 2, 3 and 4) in transcriptomic and genomic data in cetacean species located at the top of the food pyramid and heavily affected by pollutants such as heavy metals through bioaccumulation. Studies on the effects of pollutants on marine mammal species are always of great interest to readers on the subject.The paper is well written, has current and important data, and should be of great interest to the readers. The introduction and others sections provide useful information for the readers. The paper has a potential to be accepted, but some important points have to be clarified or fixed before we can proceed and a positive action can be taken.

I here summarize this points:

1- I think that the inclusion of articles on the presence of heavy metals in marine ecosystems, plant and animal organisms, seawater and sediment in the introduction will add more value to the paper. Thus, the introduction will be more holistic and comprehensive. For example

- Arisekar, U., Shakila, R. J., Shalini, R., Jeyasekaran, G., Sivaraman, B., & Surya, T. (2021). Heavy metal concentrations in the macroalgae, seagrasses, mangroves, and crabs collected from the Tuticorin coast (Hare Island), Gulf of Mannar, South India. Marine Pollution Bulletin, 163, 111971.

- Liu, R., Jiang, W., Li, F., Pan, Y., Wang, C., & Tian, H. (2021). Occurrence, partition, and risk of seven heavy metals in sediments, seawater, and organisms from the eastern sea area of Shandong Peninsula, Yellow Sea, China. Journal of Environmental Management, 279, 111771.

- Yabanli, M., & Alparslan, Y. (2015). Potential health hazard assessment in terms of some heavy metals determined in demersal fishes caught in Eastern Aegean Sea. Bulletin of Environmental Contamination and Toxicology, 95, 494-498.

- Yang, G., Song, Z., Sun, X., Chen, C., Ke, S., & Zhang, J. (2020). Heavy metals of sediment cores in Dachan Bay and their responses to human activities. Marine Pollution Bulletin, 150, 110764.

2- Figure 2 is not clear. It will be better if it is rearranged.

3- A table should be created for the cetacean species included in the article. The table should include species names, target organ or Mts, reference, etc.

The content of the article in accordance with the aims of the Toxics.

The article is scientifically sufficient.

Keywords are well chosen so that the article can be found by indexes.

The literature has been adequately critical, current and internationally evaluated by the authors.

The language of the article is correct and clear.

The discussion part is quite comprehensive in the paper.

Tables and figures are well designed and necessary.

 Acceptable after minor revisions. 

Author Response

1/ “I think that the inclusion of articles on the presence of heavy metals in marine ecosystems, plant and animal organisms, seawater and sediment in the introduction will add more value to the paper. Thus, the introduction will be more holistic and comprehensive.”AUTHORS: we are sorry but we didn’t add new references on metal concentrations in marine organisms (plants), seawater and sediment, because Referee 1 recommended reducing the introduction.

2/ Figure 2 is not clear. It will be better if it is rearranged. AUTHORS: We increased the size and quality of the figures.

3/ A table should be created for the cetacean species included in the article. The table should include species names, target organs or Mts, references, etc. AUTHORS: Table 1 indicated all information suggested.

Reviewer 4 Report

The whole manuscript was well organised, but some misspellings were exists, which should be corrected, i.e. line 39. In addition, the mechanism of MT were detailed in the context, but how to elevate the effect of MT on metals detoxification, particularly the metals that less bounded by MT, could not be found in the manuscript. As authors mentioned in manuscript that MT effect could be heavily impacted by nutritive condition, diet and other ecological and physiological factors, these processes should be slightly recovered, which inevitably beneficial to environmental management.

Author Response

Any modification recommended.

Round 2

Reviewer 1 Report

toxic-2329882

In this study, the identification of metallothionein sequences in all genomic fragments (scaffold, contig, read), cDNA and transcriptomes of cetaceans available in international databases was performed.

All points raised during the review process were considered by the authors.

This version of the manuscript is interesting and contains some novelties. Therefore, it can provide insights for future research.

Reviewer 4 Report

This revision should be adequate, so that I think this manuscript could be accepted in current form.